# *MaNrtB*, a Putative Nitrate Transporter, Contributes to Stress Tolerance and Virulence in the Entomopathogenic Fungus *Metarhizium acridum*

**DOI:** 10.3390/jof11020111

**Published:** 2025-02-01

**Authors:** Jia Wang, Yuneng Zou, Yuxian Xia, Kai Jin

**Affiliations:** 1Genetic Engineering Research Center, School of Life Sciences, Chongqing University, Chongqing 401331, China; 202326021033@cqu.edu.cn (J.W.);; 2Chongqing Engineering Research Center for Fungal Insecticide, Chongqing 401331, China; 3Key Laboratory of Gene Function and Regulation Technologies Under Chongqing Municipal Education Commission, Chongqing 401331, China; 4National Engineering Research Center of Microbial Pesticides, Chongqing 401331, China

**Keywords:** entomopathogenic fungi, *Metarhizium acridum*, *MaNrtB*, stress tolerance, virulence

## Abstract

Nitrogen is an essential nutrient that frequently determines the growth rate of fungi. Nitrate transporter proteins (Nrts) play a crucial role in the cellular absorption of nitrate from the environment. Entomopathogenic fungi (EPF) have shown their potential in the biological control of pests. Thus, comprehending the mechanisms that govern the pathogenicity and stress tolerance of EPF is helpful in improving the effectiveness and practical application of these fungal biocontrol agents. In this study, we utilized homologous recombination to create *MaNrtB* deletion mutants and complementation strains. We systematically investigated the biological functions of the nitrate transporter protein gene *MaNrtB* in *M. acridum*. Our findings revealed that the disruption of *MaNrtB* resulted in delayed conidial germination without affecting conidial production. Stress tolerance assays demonstrated that the *MaNrtB* disruption strain was more vulnerable to UV-B irradiation, hyperosmotic stress, and cell wall disturbing agents, yet it exhibited increased heat resistance compared to the wild-type strain. Bioassays on the locust *Locusta migratoria manilensis* showed that the disruption of *MaNrtB* impaired the fungal virulence owing to the reduced appressorium formation on the insect cuticle and the attenuated growth in the locust hemolymph. These findings provide new perspectives for understanding the pathogenesis of EPF.

## 1. Introduction

Numerous insect pests are prevalent in the natural environment, presenting significant challenges to agriculture, ecological balance, and human well-being [1]. Entomopathogenic fungi (EPF), a group of fungi that can cause infections and death in insects and other arthropods, have emerged as a promising and eco-friendly alternative to chemical insecticides and are extensively utilized as biocontrol agents against insect pests [2]. Well-known EPFs, such as *Metarhizium acridum* and *Beauveria bassiana*, have shown their potential and advantages in the biological control of pests within agricultural ecosystems [3]. Therefore, understanding the mechanisms that govern the pathogenicity and stress tolerance of EPF is helpful in improving the effectiveness and practical application of these fungal biocontrol agents.

In fungi, detecting and effectively utilizing nutrients are crucial for the successful establishment and rapid expansion of fungal colonies, and nitrogen, a critical nutrient, often dictates the growth rate of fungi [4]. The metabolism of nitrogen sources encompasses a range of biochemical pathways and mechanisms, such as the assimilation of ammonia, the reduction of nitrates, the fixation of nitrogen, the synthesis of amino acids, and so on [5]. Within this metabolic network, nitrate metabolism is an important part of nitrogen source metabolism in fungi.

Nitrate transporter proteins (Nrts) are pivotal in the cellular uptake of nitrate from the surroundings, playing a critical role in the assimilation, metabolism, and utilization of nitrate. The genetic exploration of these transporters in eukaryotes began in 1991 with the discovery of the *CrnA* gene, which encodes a transmembrane protein, in *Aspergillus nidulans* [6]. The *CrnA*-deficient strains exhibited nitrate uptake defects in both the conidial and the mycelial growth phases [7]. These high-affinity transport systems are part of the nitrate/nitrite porter family, a unique subset within the major facilitator superfamily (MFS), the largest group of secondary transporters [8]. Most eukaryotic organisms possess multiple Nrt protein variants, each encoded by distinct genes [9]. For instance, in the yeast *Hansenula polymorpha* (syn *Ogataea angusta*), the *YNT1* mutants exhibits a deficiency in the uptake of nitrate [10]. In the pathogenic fungus *Fusarium oxysporum* f. sp. *Lycopersici*, the targeted deletion of the nitrate-transporting gene *Ntr1* significantly impairs fungal growth when nitrate serves as the nitrogen source [11]. Similarly, in *Ustilago maydis* (syn *Mycosarcoma maydis*), the growth of nitrate transporter protein mutants is hindered in media where nitrate is the sole nitrogen source and their pathogenicity is diminished [12]. *Trichoderma atroviride*, a prevalent plant growth-promoting fungus, has been shown to rely on the nitrate transporter CHL1 for nitrogen uptake in MS medium supplemented with NH_4_NO_3_ [13]. Collectively, these findings underscore the influence of nitrate transporter proteins on fungal growth and pathogenicity. Despite their importance, the characterization of nitrate transporter genes in entomopathogenic fungi remains unexplored. Here, we investigated some biological processes that may be affected by the activity of a putative nitrate transporter, MaNrtB, in *M. acridum*.

## 2. Materials and Methods

### 2.1. Strains and Culture Conditions

The wild-type strain of *M. acridum* (WT) has been archived at the China General Microbiological Culture Collection Center (CGMCC) with the accession number 0877. In conducting the genetic engineering and fungal transformation for this study, we employed *Escherichia coli* DH5α (Solarbio, Beijing, China) and *Agrobacterium tumefaciens* AGL1 (Solarbio, Beijing, China), respectively. The fungal strains employed in this investigation were cultivated on a modified Sabouraud dextrose agar medium, specifically a one-quarter-strength formulation (1/4 SDAY). This medium is composed of 1% dextrose, 0.25% mycological peptone, 0.5% yeast extract, and 2% agar, measured by the weight to volume ratio. The cultures were maintained at a temperature of 28 °C for a period of 15 days to ensure the conidia reached maturity [14].

### 2.2. Bioinformatic Analyses

All the protein sequences were sourced from the NCBI genome database, accessible at https://www.ncbi.nlm.nih.gov/ (accessed on 25 December 2024). The specific nucleic acid sequence (MAC_03189) and protein sequence (XP_007809529.2) were obtained in FASTA format. Utilizing these protein sequences, a selection of diverse filamentous and non-filamentous fungi were made for BLASTp sequence alignment to identify similarities. Protein 3D structure maps were retrieved from the AlphaFold Protein Structure Database at https://alphafold.ebi.ac.uk/ (accessed on 25 December 2024). For constructing the phylogenetic tree, MEGA v7.0 was employed with the neighbor-joining approach. The sequences were aligned using the ClustalW algorithm integrated within MEGA and employing a bootstrap test with 1000 replicates. Another alignment parameter was set to default settings. The aligned sequences were then exported to GeneDoc 2.7 for homology analysis. Domain prediction was conducted using the SMART tool, available at http://smart.embl.de/ (accessed on 25 December 2024), and TMH prediction was performed using TMHMM at https://services.healthtech.dtu.dk/ (accessed on 25 December 2024). Additionally, BioXM v2.7.1 software was used to calculate the molecular weights and isoelectric points of the proteins.

### 2.3. Fungal Mutant Generation

The *MaNrtB* nucleic acid sequence (2535 bp) and an additional 2000 bp flanking the *MaNrtB* gene on both sides were obtained from the NCBI database. Among them, a 1.3 kb (left border) and a 1.4 kb (right border) flanking sequence of the *MaNrtB* gene locus were amplified using the primer pairs of *MaNrtB*-LF/*MaNrtB*-LR and *MaNrtB*-RF/*MaNrtB*-RR, respectively (Appendix A). The fragments were then successively ligated into the *Hin*dIII/*Xba*I and *Eco*RV/*Eco*RI-restricted pK2-PB vectors and verified by PCR using the primers LF/Pt-R and Bar-F/RR [15], respectively. Fungal transformation and validation of the transformants followed the methods described previously [16]. Potential Δ*MaNrtB* mutants were identified on Czapek–Dox (CZA) medium supplemented with 500 μg/mL glufosinate ammonium (Sigma, St. Louis, MO, USA), and the presence of the desired recombinant sequences was confirmed by PCR using the primer pairs of NrtB-VF/Pt-R and NrtB-VR/Bar-F. The pK2-sur vector was employed to generate the *MaNrtB* complemented strains (CPs). Initially, the full-length *eGFP* and *TtrpC* genes were cloned and ligated into the pK2-sur vector to create the pK2-eGFP-sur vector. Then, the promoter and ORF fragments were amplified from the WT genome using the primers NrtB-CP-F and NrtB-CP-R, and ligated to the *Hin*dIII/*Bam*HI-restricted pK2-eGFP-sur vector by one-step cloning to create the pK2-sur-*MaNrtB* construct [17], which was introduced into the Δ*MaNrtB* strain. Candidate CP strains were selected on CZA medium with 20 μg/mL chlorimuron ethyl (Sigma, Bellefonte, PA, USA), and recombinant events were verified by PCR using the primer pair of NrtB-CP-VF/GFP-VR. Reverse transcription quantitative PCR (RT-qPCR) analysis was performed to confirm the Δ*MaNrtB* and CP strains. The schematic diagrams of the pK2-PB and pK2-sur vectors are shown in Appendix A. A complete list of the primers utilized in this study is provided in Appendix A.

### 2.4. Conidial Germination and Conidial Yield Assays

Conidial suspensions at a concentration of 1 × 10^7^ conidia/mL were prepared with mature and freshly harvested conidia and a 0.05% Tween-80 solution. An aliquot of 50 μL was uniformly spread onto 1/4 SDAY plates. Over a period of 10 h, at 2-h intervals, three 1 cm^2^ sections were excised from each strain’s plate. The number of germinated conidia was counted for every 100 conidia observed. Germination was defined as the emergence of a bud whose length was more than one-third of the width of the conidium. The germination rate was determined by the following formula: (number of germinated conidia/total number of conidia) × 100%. For assessing the conidial yield, 2 μL of the conidial suspension (1 × 10^6^ conidia/mL) was deposited into 24-well plates with 1 mL of 1/4 SDAY medium in each plate well. Every three days, the yield of conidia was measured. Briefly, the fungal cultures were transferred to a 2 mL centrifuge tube. After adding 1 mL of 0.05% Tween-80 solution for grinding, the mixture was vortexed. The suspension was then diluted, and the conidial concentration was determined using a hemocytometer. Each experiment was independently performed a minimum of three times to ensure reliability.

### 2.5. Stress Tolerance Assays

The 1/4 SDAY medium was amended with 0.5 M NaCl, 1 M sorbitol (SOR), 500 μg/mL Congo red (CR), 0.01% *w*/*v* sodium dodecyl sulfate (SDS), 50 μg/mL Calcofluor white (CFW), and 60 mmol/L H_2_O_2_ to simulate adverse conditions. A 2 µL aliquot of conidial suspension (1 × 10^6^ conidia/mL) from each strain was spread onto the modified culture plates and incubated at 28 °C for 6 days. The colony growth was then photographed and measured. To evaluate the heat-shock sensitivity of the fungal strains, the conidial suspensions (1 × 10^7^ conidia/mL) were subjected to a water bath at 45 °C for durations of 2, 4, 6, and 8 h. Post-treatment, the suspensions were spread onto 1/4 SDAY medium and incubated at 28 °C for 20 h to determine the germination rates. For the UV-B tolerance of the fungal strains, the conidial suspensions (1 × 10^7^ conidia/mL) were spread on 1/4 SDAY medium and exposed to UV-B radiation at a dose of 1350 mW/m^2^ for periods of 0.5, 1.0, 1.5, and 2.0 h. Following exposure, the cultures were incubated at 28 °C for 20 h to assess the conidial germination. Each experiment was conducted in triplicate.

### 2.6. Bioassays

Two distinct bioassay methods were employed as previously described [15]. Briefly, in the topical inoculation approach, suspensions of the WT, Δ*MaNrtB*, and CP strains, prepared in paraffin oil at a concentration of 1 × 10^7^ conidia/mL, were applied to the fifth-instar nymphs of locusts, *Locusta migratoria manilensis* (5 μL per nymph). For the intrahemocoel injection method, the conidia from the fungal strains were suspended in double-distilled water (ddH_2_O) at a concentration of 1 × 10^6^ conidia/mL and injected directly into the locust hemocoel (5 μL per nymph). For both inoculation techniques, each experimental group contained 20 fifth-instar nymphs, and the negative control groups were treated with pure paraffin oil and ddH_2_O, respectively. The survival rates of the locusts were monitored and documented at intervals of 12 h throughout the experiment.

To investigate the nutrient utilization efficiency of the various fungal strains, we employed liquid CZA medium along with two modified versions of CZA, SPM and TPM, to assess the fungal development under distinct nutritional conditions. The CZA medium was supplemented with 3% sucrose and 0.3% NaNO_3_, while the SPM contained 3% sucrose and 0.5% peptone, and the TPM was formulated with 3% trehalose and 0.5% peptone. Among the three media, CZA has nitrate as the main nitrogen source donor and thus can be used to assess the nitrate transport activity of NrtBp in *M. acridum*. TPM is used to mimic insect hemolymph, as described previously [18]. These media were designed to replicate and evaluate the growth of fungi in a variety of nutrient-rich environments. Briefly, a 30 μL sample of a conidial suspension at a concentration of 10^7^ conidia/mL was introduced into each 30 mL medium, which was then incubated for 60 h at 28 °C with a shaker speed of 200 rpm to determine the biomass. This procedure was conducted in triplicate for each condition. To evaluate the host immune response, the total RNA was extracted from the fat bodies in the nymphs of locusts after injection to assess the expression levels of two key antimicrobial peptide (AMP) genes, *Attacin* and *Defensin*, using the protocol outlined in a previous study [19]. Each bioassay experiment was repeated three times.

### 2.7. Appressorium Formation Assays

Conidial suspensions for each strain were prepared in sterilized ddH_2_O to a density of 1 × 10^7^ conidia/mL. The sterilized locust hindwings were first coated with the conidial suspensions and then carefully spread onto pristine microscope slides for incubation at a controlled temperature of 28 °C. The conidia germination rate was monitored and recorded every two hours. Additionally, after a 12 h incubation period, the appressorium formation rate was also counted. This experimental procedure was conducted in triplicate to ensure reliability.

To assess the fungal penetration capability, we placed sterilized locust hindwings on 1/4 SDAY plates. A 2 µL aliquot of conidial suspension from each fungal strain, diluted to a concentration of 1 × 10^6^ conidia/mL, was dropped on the wings carefully. These were then incubated at 28 °C for 44 h. Following photography and documentation, the wings were removed to allow continued incubation for an additional four days, with photographs taken to document the colony morphology. As a control, conidial suspensions were also directly inoculated onto 1/4 SDAY medium and incubated at 28 °C for a duration of 6 days. Each treatment contained 10 wings and these assays were repeated three times.

### 2.8. Data Analyses

A one-way ANOVA test was conducted to evaluate the phenotypic estimates, followed by Tukey’s honestly significant difference (HSD) test to ascertain the significance of any observed differences among the groups.

## 3. Results

### 3.1. Features of MaNrtB and Mutant Generation

Bioinformatics analysis revealed that there is only one putative Nrt gene, designated as *MaNrtB*, in *M. acridum*. The *MaNrtB* gene spans 1612 base pairs and encodes a protein of 502 amino acids, distributed across two exons. The calculated molecular weight for the *MaNrtB* protein is 53.75 kDa, with an isoelectric point of 7.8. Domain analysis, conducted via the SMART online tool, indicated the presence of an MFS domain. As with other reported Nrt homologues [6,11,12,20,21,22], the tertiary structure of NrtB in *M. acridum* putatively contains 12 transmembrane-spanning alpha helices (TMHs)that are colored in blue with a less conserved intracellular central loop (Figure 1A and Appendix A). The sequence alignment of the orthologous NrtB proteins revealed that these TMH domains are highly conserved, whereas the central loop region exhibits a lack of order (Appendix A). Additionally, phylogenetic analysis grouped *MaNrtB* together with several of its homologs that have been evidenced to possess nitrate transport capacity, such as *A. nidulans* [6] and *F. oxysporum* [11] (Figure 1B), and these domains are notably conserved in filamentous fungi and exhibit lower conservation in yeast (Figure 1B). Overall, the significant similarity in the tertiary structures of these proteins suggests that they likely have similar functions. *MaNrtB*-knockout transformants were generated through homologous recombination. The CP strain was derived using the ectopic strategy (Figure 2B). The Δ*MaNrtB* and CP strains were validated by PCR (Appendix A). Furthermore, the expression levels of the *MaNrtB* gene in the WT, Δ*MaNrtB*, and CP strains were quantified using RT-qPCR. The expression levels of *MaNrtB* in the WT and CP strains were markedly higher than in the Δ*MaNrtB* strain, confirming the successful generation of both the Δ*MaNrtB* and CP strains (Figure 2C).

### 3.2. Disruption of MaNrtB Impaired Conidial Germination but Not Conidial Yield

Our initial assessment focused on the germination rates and conidial yields of the different fungal strains. It was observed that the Δ*MaNrtB* strain exhibited a considerably reduced germination rate compared to both the WT and CP strains after 4 h of incubation (Figure 3A). Furthermore, the Δ*MaNrtB* strain required a significantly longer time to reach 50% germination (GT_50_) (Figure 3B). In contrast, the conidial yield among the three strains was found to be statistically indistinguishable (Figure 3C). Collectively, these findings suggest that the disruption of the *MaNrtB* gene in *M. acridum* led to the retardation of conidial germination without impacting the conidial production.

### 3.3. Disruption of MaNrtB Affected Fungal Stress Tolerances

The ability to withstand environmental stress is crucial for EPF. Thus, we examined the UV-B and heat-shock stress tolerances of the WT, Δ*MaNrtB*, and CP strains. Post-UV-B irradiation, the Δ*MaNrtB* strain exhibited a markedly reduced conidial germination rate after one hour and a significantly lower GT_50_ compared to the WT and CP strains (Figure 4A,B). Conversely, under heat-shock conditions, the Δ*MaNrtB* strain displayed a higher germination rate after four hours of treatment, with a notably elevated GT_50_ (Figure 4C,D). These findings suggest that the disruption of *MaNrtB* in *M. acridum* compromised its UV-B resistance but enhanced its heat tolerance. Furthermore, spot assays revealed that the Δ*MaNrtB* strain formed smaller colonies relative to the WT and CP strains on 1/4 SDAY medium supplemented with SOR, NaCl, H_2_O_2_, CR, and CFW (Figure 5), indicating that the Δ*MaNrtB* strain was more susceptible to hyperosmotic, oxidative, and cell wall stresses.

### 3.4. Disruption of MaNrtB Attenuated Fungal Virulence

To assess the role of *MaNrtB* in virulence, we conducted bioassays using two infection methods, topical application and direct intrahemocoel injection. With topical application of the WT or CP strains, all the locusts succumbed by 8.5 days post-inoculation (dpi), whereas those inoculated with the Δ*MaNrtB* strain died after 10.0 dpi (Figure 6A). The median lethal time (LT_50_) for the Δ*MaNrtB* strain was 6.26 ± 0.80 days, which was significantly longer than the 4.90 ± 0.69 days for the WT and 5.32 ± 0.73 days for the CP strain (*p* < 0.05; Figure 6B). Similarly, when using the intrahemocoel injection method, the survival rates of the locusts infected with the Δ*MaNrtB* strain were also significantly higher than those infected with the WT or CP strains (Figure 6C). Likewise, the LT_50_ for the Δ*MaNrtB* strain, at 4.64 ± 0.67 days, was significantly longer than the 3.55 ± 0.56 days for the WT and 3.97 ± 0.60 days for the CP strains (*p* < 0.05; Figure 6D). Collectively, these findings indicate that *MaNrtB* plays a significant role in the virulence of *M. acridum*.

To explore whether the *MaNrtB* deletion impacts fungal penetration, we inoculated *M. acridum* on the hind wings of locusts and analyzed the penetration ability of each strain. The results showed that compared to the WT and CP, the Δ*MaNrtB* strain formed the smallest colony with the lowest penetration ability (Figure 7A). To elucidate the functions of *MaNrtB* in the infectious process, we analyzed the germination of conidia and the formation of appressoria on the locust hind wings for the WT, Δ*MaNrtB*, and CP strains. The absence of *MaNrtB* led to slower conidial germination on the locust hind wings (Figure 7B,C). Microscopic observation showed that all three strains formed normal appressorium, but the *MaNrtB* mutation also resulted in significantly decreased appressorium formation (*p* < 0.01; Appendix A and Figure 7D). After a 24 h induction on locust wings, the Δ*MaNrtB* strain formed about 30% normal appressoria, in contrast to the over 50% observed in both the WT and CP strains (Figure 7D).

To verify whether the *MaNrtB* deletion impacts fungal colonization in hemolymph, we first simulated fungal growth in different media, then quantified the biomass, and analyzed the uptake of carbon and nitrogen sources. It was found that the fungi accumulated the most biomass in SPM medium containing sucrose and peptone, which were the preferable choice for both nitrogen and carbon sources, and the Δ*MaNrtB* mutant accumulated about 22% biomass less than the WT (Figure 8A). Replacing sucrose with trehalose widened the gap to about 43% in TPM (Figure 8A). In the nutrient-poor CZA standard medium, the gap in accumulated biomass reached 56% (Figure 8A). Furthermore, to probe whether the *MaNrtB* gene contributed to the ability of the fungus to evade host immune defenses, the expression levels of *Defensin* and *Attacin*, Toll- and/or Imd-activating antimicrobial peptide genes involved in fungal recognition and immune response in the insect fat bodies were analyzed after injection with conidial suspension from different fungal strains. The results displayed that lower levels of expression of *Attacin* (32%) and *Defensin* (43%) were found in the locusts infected by the *MaNrtB* strain compared to those infected by the WT after 24 dpi (Figure 8B).

## 4. Discussion

NrtB, a conserved nitrate transporter in filamentous fungi, has been the subject of limited research regarding its molecular mechanisms and their relation to fungal growth. In our study, we employed homologous recombination to generate *MaNrtB* deletion and complementation transformants, conducting a thorough analysis of the *MaNrtB* functions. Our results indicated that *MaNrtB*, the nitrate transporter gene, plays important roles in the conidial germination, stress tolerances, and virulence of the entomopathogenic fungus *M. acridum*. These findings provide new perspectives for understanding the biological properties and pathogenesis of EPF.

Nitrogen sources play a pivotal role in the morphological development, production of secondary metabolites, and virulence of EPF [23]. Typically, fungi prioritize certain nitrogen sources, such as ammonium, and repress the assimilation of others, such as nitrate, a phenomenon termed nitrogen catabolite repression (NCR) [24]. In Ascomycetes, including *A. nidulans*, the key regulators AreA and AreB govern nitrogen metabolism, with nitrate metabolism gene activation primarily under the control of the transcription factors AreA and NirA [25,26,27]. AreA acts as a positive regulator that can alleviate NCR, while NirA is a transcription factor specific to the nitrate assimilation pathway, induced by nitrate or nitrite [28]. NmrA is another central regulator within the NCR pathway, modulating fungal nitrogen metabolism by interacting with and suppressing the activity of AreA and AreB [29,30,31,32]. In our previous research, we established a library of differentially expressed genes responsive to nitrate [33]. Among the genes, *MaNCP1*, a C2H2-type transcription factor gene, was identified to modulate nitrate metabolism by interacting with the *MaNmrA* in *M. acridum* [34,35]. *MaNCP1* influences nitrate metabolism and impacts nitric oxide (NO) synthesis [34], which can engage in various signal transduction pathways, leading to phenotypic alterations in the fungus [36]. Nitrate can trigger the upregulation of *MaNCP1* and *MaNmrA*, and the expression of the nitrate transporter gene *MaNrtB* is downregulated following the deletion of *MaNCP1* or *MaNmrA* [34]. This suggests reduced transport of nitrate from the extracellular environment into the cell, yet the nitrate content in the Δ*MaNCP1* and Δ*MaNmrA* strains was significantly higher than in the wild-type strain, indicating a severe disruption in nitrate catabolism [34]. As a key component of the conserved nitrate assimilation pathway, Nrts regulate cellular nitrogen utilization and further influence fungal growth and development [37]. In our research, the Δ*MaNrtB* strain displayed slower growth, reduced stress resistance, and decreased virulence against locusts, just like the mutant of Δ*MaNCP1* [38] and Δ*MaNmrA* strains [39], underscoring the important role of nitrate metabolism in sustaining fungal growth.

The capacity of conidia to withstand environmental stress is critical for the efficacy of EPF in the field, underscoring the importance of identifying robust strains for practical applications [40]. Our research revealed that the Δ*MaNrtB* strain showed reduced tolerances to UV-B and cell wall stress but enhanced resistance to heat shock. The fungal cell wall is the primary defense against external stressors [41]. Compounds such as CFW and CR are commonly utilized to challenge fungal cell walls in vitro [42]. CFW interferes with chitin assembly, and CR inhibits β-glucan synthesis, both leading to cell wall damage and triggering cell wall stress responses [43]. In *Arabidopsis thaliana*, a co-expression analysis of cell wall remodeling genes with nitrate and ammonium transporters revealed that there is a notably strong correlation between the regulation of cell wall remodeling and nitrate assimilation processes, while ammonium transporters appeared to have a more limited role in these co-expression networks [44]. Thus, the inhibited growth of Δ*MaNrtB* on 1/4 SDAY medium containing CR and CFW suggests that *MaNrtB* also plays a role in maintaining cell wall integrity.

The conidial germination rate of the Δ*MaNrtB* strain was diminished, echoing observations in *U. maydis* [12] and *F. oxysporum* [11]. However, the production of conidia remained unaffected in the Δ*MaNrtB* strain, suggesting that the *MaNrtB* gene is pivotal for conidial germination but exerts minimal influence on the process of conidiation. In *F. fujikuroi*, a reduction in biomass was noted in mutants of the nitrate transporter protein compared to the wild type [45]. Consistent with prior fungal studies, the growth rate of *F. graminearum* is dependent on the availability and type of nitrogen sources [46]. In *B. bassiana*, deprivation of nitrogen nutrients leads to reduced conidium production [47]. Disrupting the gene encoding the nitrate-responsive transcription factor NirA in *B. bassiana* causes reduced growth rates and diminished efficiency in utilizing nitrate and urea [48]. This supports the notion that the highly conserved nitrate assimilation pathway is crucial for promoting growth across various filamentous fungi.

Elucidating the impact of nitrogen sources on fungal virulence is helpful in developing novel pest control strategies. A successful infection hinges on the ability of a pathogen to overcome host defenses and secure vital nutrients necessary to complete its life cycle [49]. The nitrate assimilation pathway plays a significant role in fungal virulence. In *Colletotrichum acutatum*, a strain with a deletion of the *NirA1* gene could not grow on media using nitrate or nitrite as the sole nitrogen source and displayed reduced virulence on strawberry leaves [50]. In *N. crassa*, growth tests with nitrate as the sole nitrogen source showed that the homologous protein *nit-10* mutant was unable to grow with a low nitrate concentration [51]. In the plant pathogen *Magnaporthe oryzae*, the trehalose-6-phosphate synthase enzyme (Tps1) links glucose-6-phosphate metabolism to nitrogen source utilization, modulating nitrate reductase activity [52]. Moreover, a study showed that there is a genetic switch that consists of Tps1 and the nitrogen metabolite repressor gene *NMR1* [53]. This genetic switch controls the expression of certain nitrogen-utilization GATA-factor transcriptional factors downstream, including ASD4 (the homologous protein to AreB in *M. oryzae*), and it is necessary for regulation of a set of genes that are expressed during appressorium-mediated infection in *M. oryzae* [53]. Further research has reported that targeted deletion of *Asd4* led to an increase in glutamine levels in *M. oryzae*, which can further activate the TOR pathway and promote autophagy, thereby promoting hyphal growth while impeding appressorium formation [54,55]. The phenotypic observations of the Δ*MaAreB* strain [55] were consistent with the *M. oryzae ASD4* mutant [56]. In our study, the disruption of the *MaNrtB* gene led to delayed conidial germination, reduced appressorium formation, and attenuated growth within the insect host; thus, we speculate that the disorder of the nitrate assimilation pathways may have led to changes in the glutamine levels in vivo and influenced the TOR pathway, ultimately resulting in impaired growth and virulence in *M. acridum*.

For entomopathogenic fungi, the capacity to penetrate the insect cuticle and proliferate in the hemolymph is crucial for fungal virulence [57]. Fungi not only need to overcome the host immune response but also compete for nutrients with other microorganisms within the host; for instance, *M. rileyi* generates compounds that inhibit the proliferation of competing microorganisms, thereby facilitating its own swift expansion and contributing to the demise of the host [58]. In our study, media containing different carbon and nitrogen sources were used to simulate the growth pattern of fungi facing different nutrient environments. Correspondingly, the Δ*MaNrtB* strain showed significantly decreased total biomass in each media compared to the WT and CP strains, suggesting that the deletion of *MaNrtB* contributes to trehalose utilization, which would further affect fungal growth and dimorphic transformation within the host haemocoele, and hence fungal virulence [56]. In the nutrient-poor CZA standard medium, the gap in accumulated biomass reached 56%, while the absence of Nrts greatly limited fungal growth in nitrate-containing environments, which is consistent with previous studies [7,10,11,12]. Furthermore, fungal cells must overcome the host immune activity to propagate by yeast-like budding in the host haemocoele [59]. We assessed the immune response of locusts by quantifying the expression levels of two AMP genes, *Attacin* and *Defensin*. In this study, the locusts infected by Δ*MaNrtB* displayed decreased AMP gene expression, implicating an impaired ability of the Δ*MaNrtB* mutant in the fungal evasion from locust immunity defense. Given that the mutant strain exhibits a reduced growth rate in media with varying nutrient compositions, this could potentially impair its development under diverse conditions. Such developmental constraints might, in turn, account for the insect’s diminished defensive response.

In summary, our study sheds light on the multifaceted functions of the *NrtB* gene in the entomopathogenic fungus *M. acridum*. We have demonstrated that NrtB plays a pivotal role in multi-stress tolerance, nutrient assimilation, and fungal pathogenicity. These insights contribute to a deeper understanding of the molecular mechanisms underlying fungal virulence.

## Figures and Tables

**Figure 1 jof-11-00111-f001:**
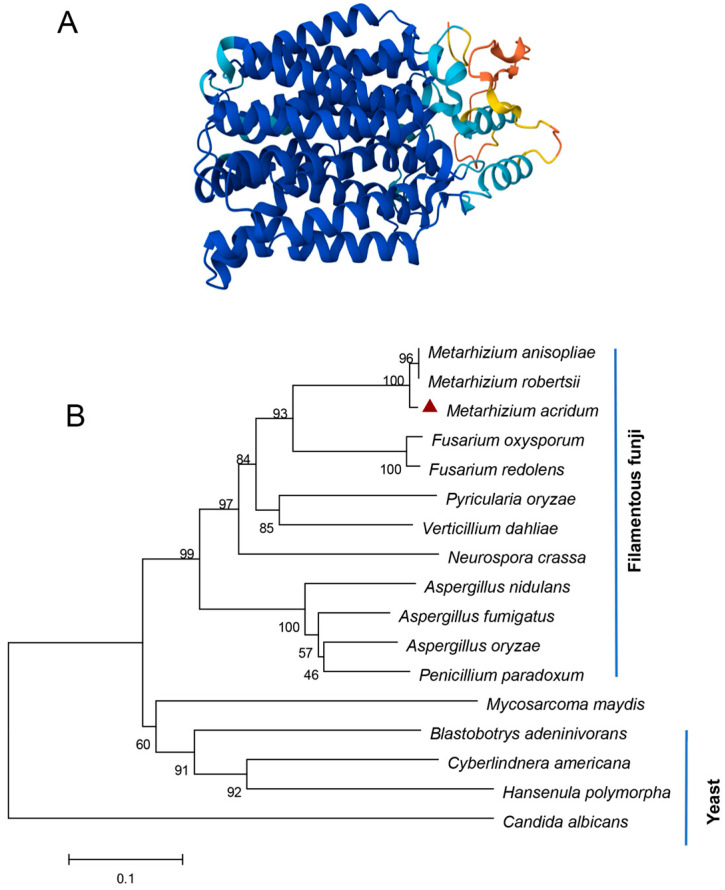
Bioinformatics of *MaNrtB*. (**A**) Domain analysis of *MaNrtB*, which contains 12 transmembrane (TM) spanning alpha helices marked in a blue color. (**B**) A phylogenetic tree was constructed with the NrtB protein sequences of *Metarhizium acridum* (EFY90826.1), *Metarhizium anisopliae* (KAF5125400.1), *Metarhizium robertsii* (XP_007820472.2), *Fusarium oxysporum* (XP_018232784.1), *Fusarium redolens* (XP_046056618.1), *Pyricularia oryzae* (XP_003710887.1), *Verticillium dahlia* (KAF3360447.1), *Neurospora crassa* (XP_957430.2), *Aspergillus nidulans* (XP_658612.1), *Aspergillus fumigatus* (KAH1494629.1), *Aspergillus oryzae* (EIT78908.1), *Penicillium paradoxum* (XP_057035394.1), *Mycosarcoma maydis* (XP_011390345.1), *Blastobotrys adeninivorans* (CAQ77149.1), *Cyberlindnera americana* (QFR37177.1), *Hansenula polymorpha* (XP_018213600.1), and *Candida albicans* (XP_717110.1). The red triangle represents the NrtB homologous protein in *M. acridum*.

**Figure 2 jof-11-00111-f002:**
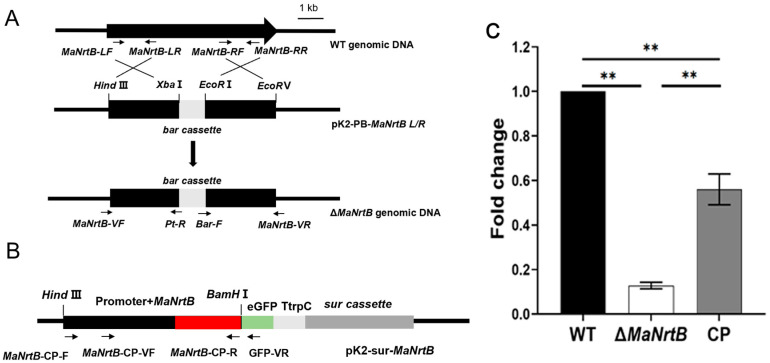
Disruption and complementation of *MaNrtB*. (**A**,**B**) Schematic diagrams of the knockout and complementation vector constructions. Black arrows indicate the positions of the primers. (**C**) Verification of transformants by RT-qPCR. Error bars = mean ± SEM. Asterisks indicate a significant difference at (**) *p* < 0.01.

**Figure 3 jof-11-00111-f003:**
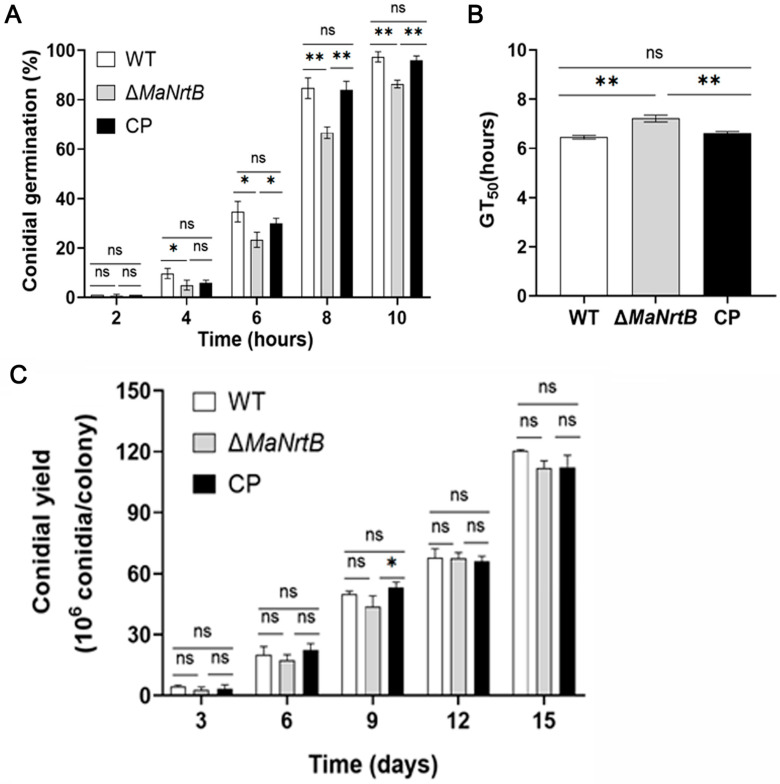
Deletion of *MaNrtB* affected the conidial germination but not the conidial yield. (**A**) The germination rate of each strain of *M. acridum* at different times. (**B**) The GT_50_ of each strain during germination. (**C**) The conidia yield of each strain on 1/4 SDAY medium for 3 d, 6 d, 9 d, 12 d, and 15 d. Error bars = mean ± SEM. Asterisks indicate a significant difference at (*) *p* < 0.05, (**) *p* < 0.01, or (ns) *p* > 0.05.

**Figure 4 jof-11-00111-f004:**
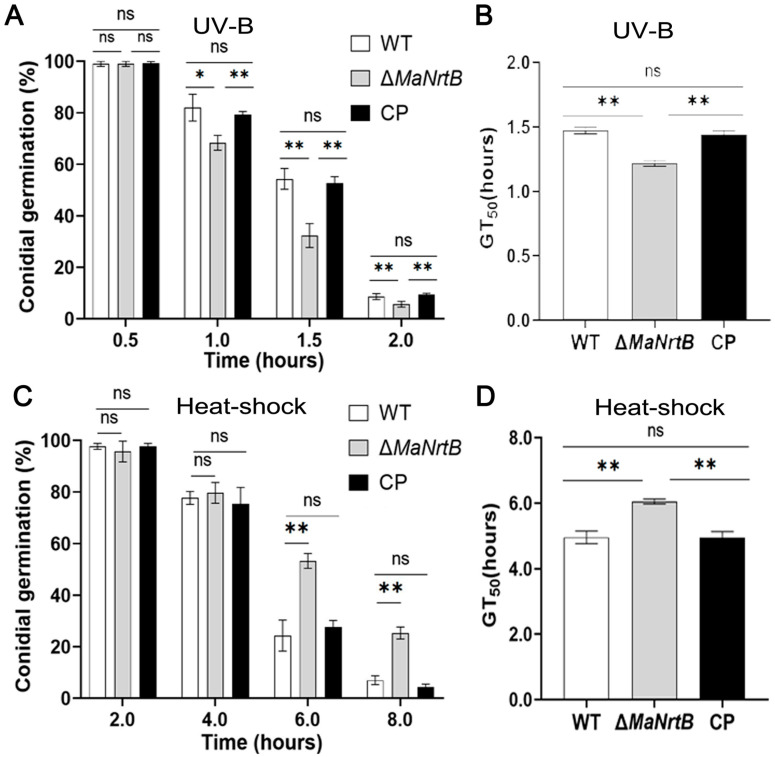
Deletion of *MaNrtB* reduced the fungal tolerance to UV-B irradiation but increased the fungal tolerance to heat shock. (**A**) Conidial germination treated with UV-B irradiation for 0.5 h, 1.0 h, 1.5 h, and 2.0 h. (**B**) The GT_50_ under UV-B irradiation. (**C**) Germination rates treated with heat shock for 2.0 h, 4.0 h, 6.0 h, and 8.0 h. (**D**) GT_50_ under heat shock. Error bars = mean ± SEM. Asterisks indicate a significant difference at (*) *p* < 0.05, (**) *p* < 0.01, or (ns) *p* > 0.05.

**Figure 5 jof-11-00111-f005:**
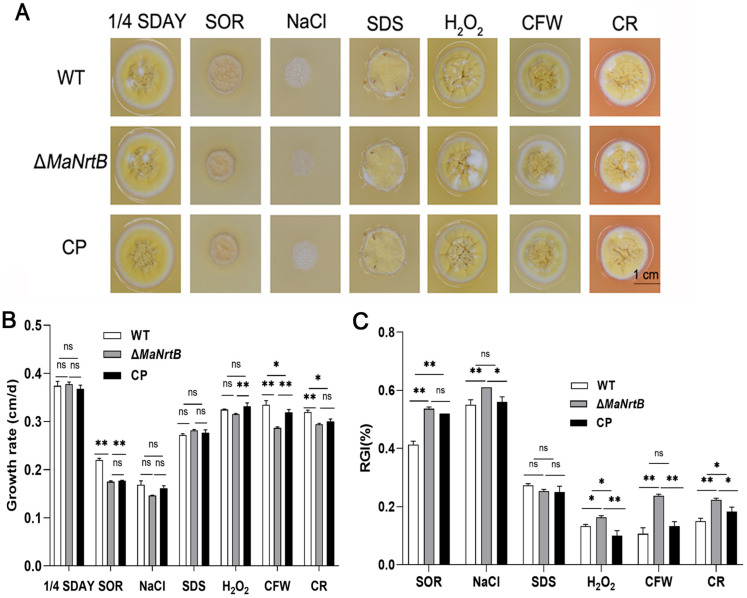
Disruption of *MaNrtB* reduced tolerances to multiple chemical reagents. (**A**) Colony morphologies of *M. acridum* strains grown for 6 days on 1/4 SDAY with different chemical reagents. (**B**) The growth rates of *M. acridum* strains under different chemical reagents. (**C**) Relative growth inhibition rates of *M. acridum* strains under different chemical reagents. Error bars = mean ± SEM. Asterisks indicate a significant difference at (*) *p* < 0.05, (**) *p* < 0.01, or (ns) *p* > 0.05.

**Figure 6 jof-11-00111-f006:**
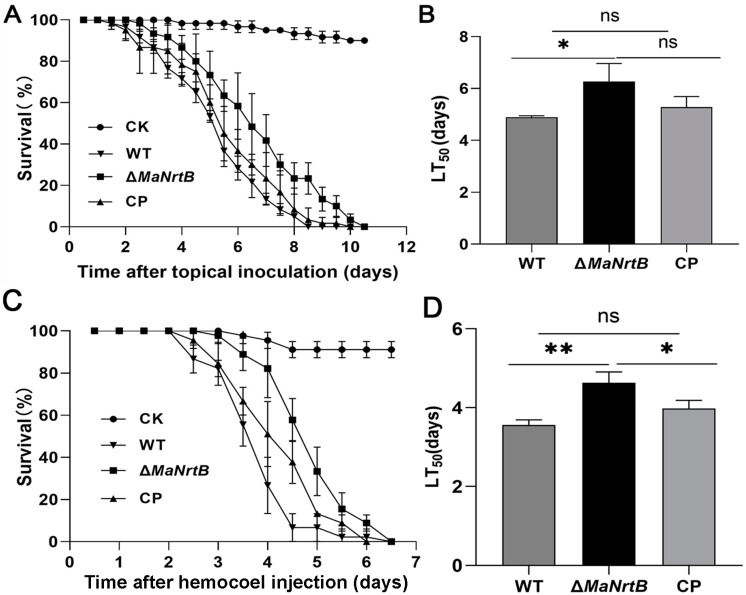
Disruption of *MaNrtB* impaired fungal virulence. (**A**) Survival rates of locusts after topical application of fungal conidia. Liquid paraffin oil as a control. (**B**) LT_50_ of fungal strains against locusts by topical application. (**C**) Survival rates of locusts after hemocoel injection of conidia. Sterile water as a control. (**D**) LT_50_ of fungal strains against locusts by hemocoel injection. Error bars = mean ± SEM. Asterisks indicate a significant difference at (*) *p* < 0.05, (**) *p* < 0.01, or (ns) *p* > 0.05.

**Figure 7 jof-11-00111-f007:**
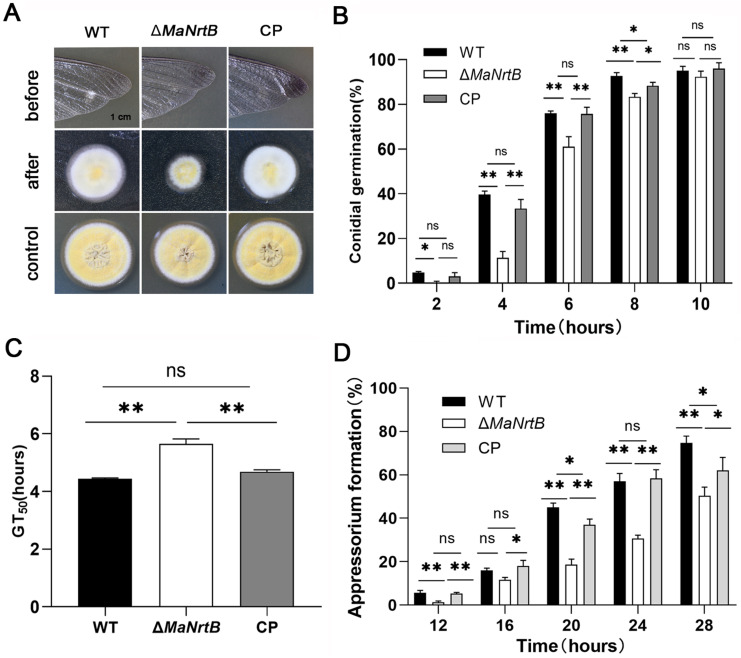
Disruption of *MaNrtB* affected locust cuticle penetration of *M. acridum*. (**A**) Penetration assays. (**B**) The germination rate of the WT, Δ*MaNrtB* and CP strains growing on the hind wings of locust for 2 h, 4 h, 6 h, 8 h and 10 h. (**C**) The GT_50_ of each strain on the hind wings of locust. (**D**) The appressorium formation rates of the WT, Δ*MaNrtB* and CP strains growing on the hind wings of locust for 12 h, 16 h, 20 h, 24 h, and 28 h. Error bars = mean ± SEM. Asterisks indicate a significant difference at (*) *p* < 0.05, (**) *p* < 0.01, or (ns) *p* > 0.05.

**Figure 8 jof-11-00111-f008:**
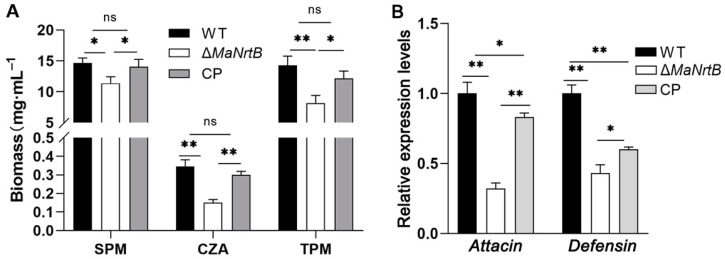
Disruption of *MaNrtB* impaired fungal growth in locust hemolymph. (**A**) The biomass levels were quantified from the 3-day-old submerged cultures in CZA and three amended media, sucrose–peptone medium (SPM) and trehalose–peptone medium (TPM). (**B**) The relative expression of *Attacin* and *Defensin* in locust fat bodies was determined at 24 h after injection by RT-qPCR. Error bars = mean ± SEM. Asterisks indicate a significant difference at (*) *p* < 0.05, (**) *p* < 0.01, or (ns) *p* > 0.05.

## Data Availability

Data are contained within the article and Appendix A.

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
