# Peer review of "MaNrtB*, a Putative Nitrate Transporter, Contributes to Stress Tolerance and Virulence in the Entomopathogenic Fungus *Metarhizium acridum"

_jof, 2025, doi:10.3390/jof11020111_

Round 1
Reviewer 1 Report
Overview and general opinion
Entomopathogenic fungi are one of the most promising alternatives for the control of various agricultural and livestock pests. Research into their biology, virulence and pathogenicity is of great value in improving their effectiveness against these pests. The authors present an investigation into the involvement of MaNrtB, a nitrate transporter, on various characteristics of M. acridum, highlighting its virulence and adaptation features.
The manuscript is well-written, with clearly structured sections. Only a few minor revisions are mentioned below.
Abstract: For better context, add that the bioassays were done on the locust Locusta migratoria manilensis.
Line 69-70: reserve this finding for the conclusions section.
Line 81-82: Please add a reference to conidia maturation, as several studies mention 21 days of maturation.
Line 89: MEGA?
Line 98: which methods?
Line 146: for how long?
Line 148: Describe what SPM and TPM mean
Line 157: nymphs of locusts.
Line 162: Did you use sterilized ddH2O alone? How did you avoid the hydrophobicity of the conidia?
Line 167-168: What control groups did you use?
Figure 6A: change “topical inoculation” by “topical application”.
Figure 6B: change “time (days)” by “time (days) after hemocoel injection”
Reviewer 2 Report
Review of the manuscript “MaNrtB, a nitrate transporter, contributes to stress tolerances and virulence in the entomopathogenic fungus Metarhzium acridum”
In this manuscript, the authors studied the involvement of the putative gene NrtB encoding a nitrate transporter in the growth, virulence and tolerance to abiotic stresses of Metarhizium acridum. The strategy was to generate a ΔNrtB null mutant and a reintegrant of the wild-type NrtB gene in the M. acridum strain, and evaluated the pathogenicity on fifth instar nymphs of Locusta migratoria manilensis and the tolerance to UV-B, high molarity stress (NaCl and sorbitol), cell wall disruptors (Congo red and Calcofluor) and oxidative stress (H2O2). They tested the penetration into Locusta wings and measured the lethality in nymphs observing that the absence of the nrtB gene delays the death of the insects. Briefly, their results suggest that the putative nitrate transporter Nrt1 is involved in germination and growth, having an impact on the time of host death.
This manuscript is interesting. It contributes to the knowledge of a gene of the entomopathogenic fungus Metarhizium acridum, which codes for a nitrate transporter. This gene is somehow connected to various cellular responses.The authors need to perform a deep bioinformatics analysis of the gene, including protein modeling to explain the protein domains. Show a detailed explanation of the complementation and null mutation constructs in vitro, and a molecular demonstration of the null mutation and complementation in vivo. The assays to study the involvement of NtrBp in germination, growth, virulence, etc. are clear, there is no doubt. In addition, it is interesting to demonstrate its biochemical function as a nitrate transporter and its localization in the membrane using a fluorescent marker, these assays will complement the work.
Material and methods
Line 88-89 fill in the parameter information for the phylogenetic tree including number of bootstrap replicates applied, etc.
Generation of fungal mutants
Line 93-101
Add detailed information on the construction of the nrtB in vitro null mutant, since from the NCBI database it used at least 6535 bp. Provide information about the construction and its molecular verification by PCR, restriction and/or sequencing, this information could be added in the supplementary material file.
Line 101-103 About the complementary construct; please add a detailed description of the construct indicating the selection marker and the name of the specific oligos used to amplify the wild-type sequence of the promoter+ORF cloned into the upstream SUR cassette and the terminator sequence cloned into the downstream SUR cassette, how many bp of each. This detailed information could be added in the supplementary material file.
Line 110, specify whether you used freshly harvested conidia.
For the conidia production assay;
Line 118, add which medium (solid or liquid) was used in the 24-well plates to measure conidia production.
Bioassays
Line 140, add how many nymphs were used with each treatment
Line 142, add how many µL were injected to each nymph and how many nymphs were used with each treatment
Lines 170-171 add how many wings were used in each treatment and number of independent assays.
Results
Line 185 How was the molecular weight of the 137.1 kDa NrtBp calculated if it is made up of 502 amino acids? The Arabidopsis thaliana Nrt crystal protein is a dimer. Is this the same for the M. acridum NrtBp? The MFS domain requires detailed analysis comparing it with previous enzymes with demonstrated activity such as the Nrt from N. crassa or others.
Figure 1A (page 5)
The protein domain scheme is simple. The MFS of the corresponding Neurospora crassa protein contains 10 transmembrane domains. M. robertsii and M. acridum contain 12 putative transmembrane domains; these transmembrane domains can be marked in Figure S1.
Replace the current scheme (1A) with protein modeling; crystals of some Nrt proteins from Arabidopsis and E. coli, among others, have been published and can be used as a reference in modeling MaNrt1Bp. It will be more interesting.
In the caption of Figure 1, add the GenBank accession number of the respective genes of B. adeninivorans, C. americana, O. polymorpha, and C. albicans. And write the complete genera of the microorganisms cited for the first time in the manuscript, not just the species.
Lines 192-194 How do you explain the difference in the expression level of the NrtB gene in the WT vs CP strains? It is 1.0 vs 0.5 and this difference is significant. Did you analyze the sequence? Do you have molecular evidence that the null mutant and the complementing strain contain only one integration in each case? And molecular evidence of the correct substitution in each case? Or do you have additional evidence that in the complementary strain the construct is complete? Is it just one copy, and did it replace the null mutation by gene replacement or was it integrated ectopically? This information and molecular evidence should be added to the manuscript or in supplementary material.
Lines 218-220 Why did you expect an impact on conidia production?
Lines 234-235 How do you explain that the absence of the NrtB gene compromises UVB resistance but improves heat tolerance in M. acridum?
Lines 257-258. The general observation from the bioassays is that virulence is attenuated. The authors wrote “…these findings indicate that MaNrtB plays an important role in virulence…” but this phenotype could be the result of decreased growth of the fungus in the host. In this condition the null mutant of the NrtB gene does not have the best fitness condition.
Lines 280-282. Please add micrographs of the appressoria of the WT, null mutant and CP strains in the supplementary material.
Discussion
It will be interesting to show a molecular model of the role of NrtBp in these networks leading to the different phenotypes observed.
Discuss how you think this knowledge can be applied to biocontrol of insect pests.
Phenotypic characterization of the M. acridum NrtB gene null mutant and the complementary strain constructed in this work is justified, in order to find the processes in which they are involved. The results obtained in the experiments performed strongly indicate that a major problem is that growth is affected. If this gene codes for a putative nitrate transporter, it is important to demonstrate its activity, at least indirectly, for example by evaluating the growth of the wild type, the null mutant and the complementary strains in a nitrate-free or low-nitrate medium, or by complementing a null mutant in another organism where the biochemical-physiological function as a nitrate transporter has been demonstrated, for example in N. crassa or another fungus, and to demonstrate its membrane localization.

Round 2
Reviewer 2 Report
In this manuscript, the authors studied the involvement of the Metarhizium acridum NrtB gene, which encodes a putative nitrate transporter, in growth, virulence and tolerance to abiotic stresses. The strategy was to generate a ΔNrtB null mutant and a wild-type NrtB gene reintegrator in the M. acridum strain, and evaluated the pathogenicity in fifth instar nymphs of Locusta migratoria manilensis and the tolerance to UV-B, high molarity stress (NaCl and sorbitol), cell wall disruptors (Congo red and Calcofluor) and oxidative stress (H2O2). They tested the penetration into the wings of Locusta and measured the lethality in nymphs observing that the absence of the nrtB gene delays the death of the insects. Briefly, their results suggest that the putative nitrate transporter Nrt1 is involved in germination and growth, having an impact on the timing of host death.This manuscript contributes to the knowledge of a gene from the entomopathogenic fungus Metarhizium acridum, which encodes a putative nitrate transporter. This gene is somehow connected to several cellular responses due to its involvement in nitrate assimilation. This is expected for any gene involved in nutrient assimilation. The authors need to perform a deep bioinformatics analysis of the gene, including the modeled protein with another previously studied one used as a reference in the modeling, preferably with crystal structure data of the reference nitrate transporter protein to explain the protein domains in the protein in question (MaNrtB). The authors performed the protein modeling but there is no discussion about its importance. I think it supports its putative activity, but considering that it is a gene of unknown function so far, it is important to support it with analyses such as nitrate transporter activity and the cellular location that would be expected in the membrane. In this new version, the in vitro construction of the null and complementary mutation is better detailed. But there is a serious problem because some of the oligos designed for both constructions do not align with the sequence of the NrtB gene (MAC_03189) or flanking sequences that include more than 6000 bp downstream and 6000 bp upstream of the NrtB gene in the NCBI database for the Metarhizium acridum genome sequence. If the sequence they based it on differs from the sequence of M. acridum ARSEF 324 and explains these anomalies, they should upload this sequence to the databases and provide the access code to verify without problem the oligos they designed. The tests to study the participation of NtrBp in germination, growth, virulence, etc. are clear, there is no doubt. But there is a doubt about the gene studied in this work. I strongly suggest carefully reviewing the molecular study, demonstrating the cellular localization of the NrtB protein and evaluating nitrate transport especially because it is a gene of unknown function, or at least there is no biochemical evidence in this entomopathogenic fungus. For the reasons mentioned above, I do not recommend its publication.
Detailed observations
MATERIAL AND METHODS
Fungal mutant generation: Lines 97-117
In supplementary Table 1, REVIEW THE SEQUENCE OF EACH PRIMER DESIGNED FOR THE NOCK OUT OF THE GENE AND FOR COMPLEMENTING IT, CHECK THERE ARE NOT ANY MISTAKE. AND ADD A COLUMN WITH THE SIZE IN BP OF EACH AMPLICON. The oligos designed to test the null mutation of the gene in the fungal transformants under study must be located outside the region selected for gene replacement by homologous recombination. Moreover were not possible to verify the alignment sites.
Line 101.
Homogenize the names of the oligos, because in Supplementary Table S1 do not begin with Ma.
Line 103. In the manuscript the authors mention that in pk2-PB were used EcoRV/EcoRI to insert the downstream fragment of nrtb, but in the cover letter they mention to use EcoRV/EcoRII …
Add the map of pk2-PB in supplementary material.
Line 104. Correct promers, should be “primers”
Line 105. Add the name of oligos used to confirm the null mutant nrtb of M. acridum. And add in supplementary material the figure of the gel electrophoresis of the amplicons observed in the mutant, absent in the wt, using a positive control of amplification like TEF gene or other gene.
Line 110. Please add the size in bp of the fragment containing the promoter and ORF of the NrtB gene and specify the terminator region used for the NrtB gene. Please add a map of the PK-sur vector in the supplementary material containing the information needed to clearly understand the poorly described construction in the manuscript.
Bioassays
Line 154-155
Should add in each technnique how many independent assays were performed.
Lines 163-164
The authors wrote” ..CZA has nitrate as the main nitrogen source donor and thus can be used to assess the nitrate transport activity of NrtBp in M. acridum..” . Formally, the authors did not show data on the measurement of nitrate transport activity; the growth is only suggestive. To investigate the function of the protein encoded by the NrtB gene as a nitrate transporter, the authors need to demonstrate that the M. acridum NrtB gene is able to complement this function in a null mutant of an organism in which the respective gene has previously been shown to be a nitrate transporter, or measure nitrate transport in the mutant and in complemented M. acridum strains.
RESULTS
Page 5 Figure 1.
The authors showed the tertiary structure of the predicted protein encoded by the M. acridum NrtB gene, but comparison with the NrtB protein of other organism and the merge of both tertiary structures is needed.
Page 7
Map of Figure 2A.
Please include the expected amplicon sizes or add these data to Table S1.
In this map (Figure 2A), the position of the oligos used to confirm the null mutant is not acceptable; the oligos NrtB-Vf and NrtB-Vr must align outside the genomic DNA sequence used to construct the in vitro deletion.
Line 210.
Check the correct size of the ORF of the MnrtB gene. Is it 1612 or 2535 bp? The information throughout the manuscript should be consistent.
Lines 214-218. Analyze the similarities between the hypothetical NrtB protein from acridum and the respective protein with known biological function in some of the organisms included in Figure S1.
Lines 221-224:
The expression level of the MaNrtB gene in the complemented strain vs. the wt strain is significant, and in the null mutant there is some expression. What do these results mean? Is it explained only by ectopic integration in the reintegrant? Do you have evidence of the copy number of integrations in the null mutant and in the complemented mutant? If so, please add the results in the supplementary material.
Does M. acridum contain only one NrtB gene or are there more Nrt genes in its genome? Did you perform this bioinformatics analysis? Please add this information at the beginning of the results, in the bioinformatics analysis.
DISCUSSION
With the putative biological function of the protein coded by the NrtB gene like a nitrate transporter, is not a surprise the effects of the null mutation on any process; growth, pathogenicity, immune response, etcetera. Considering the restriction to use organism with genetic manipulation in the pest control, which possibility do you propose in the use of the NrtB gene to improve the pest control?

Round 3
Reviewer 2 Report
The 3th review of the manuscript “MaNrtB, a putative nitrate transporter, contributes to stress tolerances and virulence in the entomopathogenic fungus Metarhzium acridum”
General coments:
The authors improved the supplementary material.
The authors have corrected some errors in the manuscript, but have not yet completed a thorough review. For example, in this version 3 there are grammatical errors, and even details such as the integration of the jof-3352404-peer-review-v3 manuscript are sloppy. It is not acceptable to start "Results" with two figures instead of a text explaining which experiments were performed to answer which hypothesis.
In the cover letter, the answer to the question of evidence for copy number of integrations in the null mutant and the complemented strain of the NrtB gene, they argued that “However, based on the genetic constructs used and standard procedures for gene transformation in our model organism, we anticipate that integration events are likely to be single copy.” However, scientific research is based on data, not assumptions, and multiplex PCR of a known single copy gene and the gene under study is a simple experiment to perform and provides information on the number of integrations.
In Introduction (see Lines 69-71) the authors wrote “Here, we systematically investigated the biological functions of the nitrate transporter protein gene MaNrtB in the M. acridum.”…
Change by “Here, we investigated some biological processes that may be affected by the activity of a putative nitrate transporter MaNrtB in M. acridum…”
This observation is due to the fact that in this manuscript the authors describe and study a gene encoding a putative nitrate transporter, so the biological function is nitrate transport. Some products of nitrate metabolism, including nitric oxide, could affect different biological processes, such as cell differentiation, and these metabolites or the enzymes or genes encoding these enzymes are not included in the experiments reported in this work.
In Results
In Figure 7: In the locusta wing, the nrtB mutant grows less than wt and CP, but how can you explain a better germination in the locusta wing compared to the ¼ SDAY medium, even from 4 h post-inoculation? Compare Figures 7B and 7C with Figures 3A and 3B.
In Fig. 8, although the mutant grows less than the wild type on the trehalose medium TPM or SPM, this is also observed on the minimal medium CZA. This suggests that the fungus, as a consequence of the damage to the nitrate transporter, has a slower growth rate, affecting its development in any condition. This would explain the insect's lower defense reaction.
The DISCUSSION is well elaborated and supported with previous reports.
But they argue that if the growth of the null mutant MaNrtB is inhibited in ¼SDYA medium containing CR and CFW, it suggests that MaNrtB gene also plays a role in maintaining cell wall integrity (see lines 389-391). Is it the NrtB transporter or is it the availability of nitrate inside the cell and/or its metabolism that is responsible for maintaining cell wall integrity? In my opinion, other types of experiments are needed to support this suggestion.
Considering the results, it is clear that nitrate uptake, as the first step in nitrate metabolism, is important for growth under any conditions, including germination and growth under saprophytic conditions and in the insect. It is important because the nitrate metabolism is necessary in the production of ammonia by action of nitrate reductase and nitrite reductase and nitrogen incorporation in glutamate and glutamine and thereof in the biosynthesis of amino acids and many others nitrogen compounds, including other molecules like NO which produces oxidative stress and participates in the induction of specific cellular differentiation processes. Thus, this work contributes to understanding the importance of a putative nitrate transporter in fungal growth under many conditions and some differentiation processes even when the fungus is confronted with its host. Future studies will shed light on the molecular mechanisms involved in these interesting processes.
In general, with a full review and correction of the manuscript, it could be published.
